# Swift and Reliable “Easy Lab” Methods for the Sensitive Molecular Detection of African Swine Fever Virus

**DOI:** 10.3390/ijms22052307

**Published:** 2021-02-25

**Authors:** Ahmed Elnagar, Jutta Pikalo, Martin Beer, Sandra Blome, Bernd Hoffmann

**Affiliations:** Friedrich-Loeffler-Institut, Institute of Diagnostic Virology, 17493 Greifswald-Insel Riems, Germany; Ahmed.Elnagar@fli.de (A.E.); Jutta.Pikalo@fli.de (J.P.); Martin.Beer@fli.de (M.B.); Sandra.Blome@fli.de (S.B.)

**Keywords:** African swine fever virus, DNA extraction, real-time PCR, easy lab

## Abstract

African swine fever (ASF) is a contagious viral hemorrhagic disease of domestic pigs and wild boars. The disease is notifiable to the World Organisation for Animal Health (OIE) and is responsible for high mortality and serious economic losses. PCR and real-time PCR (qPCR) are the OIE-recommended standard methods for the direct detection of African swine fever virus (ASFV) DNA. The aim of our work was the simplification and standardization of the molecular diagnostic workflow in the lab. For validation of this “easy lab” workflow, different sample materials from animal trials were collected and analyzed (EDTA blood, serum, oral swabs, chewing ropes, and tissue samples) to identify the optimal sample material for diagnostics in live animals. Based on our data, the EDTA blood samples or bloody tissue samples represent the best specimens for ASFV detection in the early and late phases of infection. The application of prefilled ready-to-use reagents for nucleic acid extraction or the use of a Tissue Lysis Reagent (TLR) delivers simple and reliable alternatives for the release of the ASFV nucleic acids. For the qPCR detection of ASFV, different published and commercial kits were compared. Here, a lyophilized commercial kit shows the best results mainly based on the increased template input. The good results of the “easy lab” strategy could be confirmed by the ASFV detection in field samples from wild boars collected from the 2020 ASFV outbreak in Germany. Appropriate internal control systems for extraction and PCR are key features of the “easy lab” concept and reduce the risk of false-negative and false-positive results. In addition, the use of easy-to-handle machines and software reduces training efforts and the misinterpretation of results. The PCR diagnostics based on the “easy lab” strategy can realize a high sensitivity and specificity comparable to the standard PCR methods and should be especially usable for labs with limited experiences and resources.

## 1. Introduction

African swine fever (ASF) is an OIE (World Organisation for Animal Health)-listed and devastating disease of domestic pigs and wild boars caused by a complex DNA virus of the genus *Asfivirus* in the *Asfarviridae* family [1]. The length of the African swine fever virus (ASFV) genome varies from 170 to 190 kbp among different isolates, and the number of open reading frames (ORFs) ranges from 151 to 167 [2]. In Africa, argasid ticks of the genus *Ornithodoros* can transmit the virus [3], while outside Africa, transmission via direct contact is more prevalent. ASFV can deliver very high lethality (up to 100%) in susceptible Suidae and causes significant economic losses to the pig industry [4].

ASFV is currently endemic in large parts of sub-Saharan Africa and Sardinia [5]. In 2007, the virus emerged in Georgia, and then it spread to several countries in Europe and Asia. Here, the outbreak of ASF causes a large number of deaths among domestic pigs and wild boars [6]. The typical clinical signs of ASF are high fever, rapidly deteriorating general health, respiratory distress, and hemorrhage [7]. Currently, no vaccine is available, and surveillance strategies, strict outbreak response policies, and eradication programs are the only tools to prevent the further emergence and spread of ASFV.

The common laboratory diagnostic methods for the direct detection of ASFV include virus isolation (VI), hemadsorption test (HAD), and different molecular genetic techniques, such as loop-mediated isothermal amplification (LAMP), recombinase polymerase amplification (RPA), and polymerase chain reaction (PCR). Furthermore, antigen detection can be performed by the enzyme-linked immunosorbent assay (ELISA) or fluorescent antibody tests (IFTs). However, some methods are very laborious (virus isolation) or not sensitive enough for animals with low virus levels. Antigen detection can be impaired in the presence of antibodies [8].

Therefore, conventional and real-time PCR have been considered to be reliable methods for ASFV detection [7,9] and are recommended by the OIE. In addition, PCR has been shown to be an excellent and rapid technique that can be used as a routine diagnostic tool for ASFV in either surveillance, control, or eradication program [7,9,10,11,12,13].

The objective of this study was to evaluate and validate reliable and easy molecular diagnostic methods for the so-called “easy lab” concept. Therefore, prefilled and easy-to-handle DNA extraction/releasing procedures were combined with established standard PCR procedures for the detection of ASFV. Easy lab can be defined as the simplification and standardization of the molecular diagnostic workflow in the lab aimed at realizing a high sensitivity and specificity with maximal repeatability, reproducibility, and robustness. It should be applicable for users and labs with limited facilities and resources in molecular diagnostics.

Three key points were investigated in this study. First, we identified the best sample material for accurate diagnosis of ASFV in the “easy lab” setting based on different specimens originating from different animal experiments and field samples from wild boars during the 2020 ASFV outbreak in Germany. Second, we evaluated several extraction methods for DNA isolation by comparing standard methods with different manual and automated extraction systems and other alternatives for nucleic acid release without the need to use a commercial extraction kit. Third, we tested different commercial real-time PCR kits, assays, and thermocyclers for improving the speed, sensitivity, and specificity of ASFV detection. Based on the generated data, the identification of the optimal workflow for ASFV nucleic acid detection in differently equipped and experienced labs should be supported. 

## 2. Results

### 2.1. Identification of the Best Sample Matrix for ASFV Detection

Based on three different animal experiments (1, 6, and 7), a comparison of samples was undertaken to select the best sample with regard to different matrices (EDTA blood, serum, oral swabs, and chewing ropes) and different time phases after inoculation (initial and late). The data showed that EDTA blood could detect ASFV DNA in both phases of the infection. ASFV DNA could be detected also in other matrices, but with restrictions (Figure 1A,B). Serum samples delivered comparable results to EDTA blood, but only in the later stage of infection (Figure 1B and Appendix A). In contrast, oral swab and chewing rope samples showed lower viral genome loads at all sampling dates; some even yielded negative results. Therefore, oral swabs and chewing ropes could be defined as inappropriate specimens (Appendix A). Of the bloody tissue samples, spleen showed the most reliable results with a comparative sensitivity to EDTA blood samples (Appendix A).

### 2.2. DNA Extraction Methods

To obtain a wide applicable range for viral DNA isolation, a comparison was performed between seven extraction methods (Figure 2, Table 1 and Appendix A). All methods were analyzed by the qPCR assay published by Haines et al. [13]. It could be demonstrated that all tested methods were quite sensitive, efficient, and convenient for DNA isolation from all sample materials, depicted in Figure 2. The qPCR results for sample DNA obtained by the tested extraction kits were found to be very similar in terms of Ct values. No differences could be observed between the silica membrane- and magnetic bead-based kits, the 100 and 200 µL sample starting volumes, and the non-prefilled and prefilled extraction plates. No performance differences could be observed between using the IndiMag^®^ Pathogen Kit (non-prefilled) and the IndiMag^®^ Pathogen Cartridge formats (prefilled). Furthermore, the IndiMag^®^ Pathogen IM48 Cartridge and the IndiMag^®^ Pathogen KF96 Cartridge performed equally well irrespective of the magnetic bead processing platform (KingFisher Flex and IndiMag48) used. A slightly lower sensitivity was obtained from the genome release method by virotype Tissue Lysis Reagent (TLR), whereby false-negative results were only observed for a few samples with a very low genome load (Ct value > 33, Appendix A). 

### 2.3. Rapid Amplification and ASFV Detection Using Different qPCR Assays

Using the extracted eluates of method E (IndiMag® Pathogen IM48 Cartridge), a comparison of four different qPCR assays was carried out (Figure 3 and Appendix A). The in-house Haines qPCR (Haines assay), the modified Universal Probe Library (UPL) qPCR method from the EU reference laboratory (EURL assay), and the commercial virotype ASFV 2.0 qPCR (virotype assay) were conducted on the Bio-Rad CFX96 real-time PCR cycler, whereby the commercial lyophilized IndiField ASFV PCR (IndiField assay) was applied on the IndiField thermocycler based on the matching PCR tubes. To avoid false-negative results due to PCR inhibitors or improper nucleic acid extraction, external and internal controls were co-amplified for all samples. The corresponding results of the internal controls are presented in Appendix A. 

Regarding the target detection, all tested samples amplified on the Bio-Rad CFX96 cycler produced identical qualitative positive and negative ASFV results. Furthermore, the variability of the Ct value between the three assays was very low, while the template volumes of the three assays with 2.5, 2.0, and 5.0 µL were slightly different. In comparison with the OIE-recommended modified UPL PCR assay (EURL assay), the ΔCt value was calculated for the tested samples. For the EDTA blood samples, the mean Ct values of the virotype assay and the in-house Haines assay were 1.2 and 1.1 Cts lower in comparison with the EURL method. This difference was further confirmed with the serum and oral swab samples (Table 2). 

Overall, the IndiField ASFV PCR showed the lowest Ct values and the highest sensitivity. In comparison with the EURL method, the mean Ct value for the EDTA blood samples was 3.7 cycles earlier with the IndiField PCR. Similar Ct values could be identified for the other tested matrices (Table 2). Furthermore, 10 samples with a very low viral load scored negative with the three methods performed on the Bio-Rad CFX96 cycler, but positive on the IndiField thermocycler using the IndiField ASFV PCR. Here, positive results with Ct values between 33.0 and 40.4 could be ascertained for these 10 samples (Appendix A). The improved sensitivity of the IndiField ASFV PCR is probably based on the higher template input of 20 µL.

### 2.4. Analysis of Field Samples from ASFV Outbreak in Germany 2020

A comparison was carried out between three different extraction methods (B, E, and G) and four different qPCR assays (1, 2, 3, and 4). For the amplification of the TLR-released blood samples with the IndiField ASFV PCR on the IndiField thermocycler using 20 µL template, some inhibition effects could be observed. Therefore, we diluted the template in RNase-free water with a 1:1 dilution factor (10 µL template added to 10 µL water) to reduce the concentration of PCR inhibitors. All samples extracted by the NucleoMagVet Kit and the IndiMag® Pathogen Kit were detected positive. Only 12 out of 14 samples extracted with the virotype TLR method were detected positive (2 out of 14 samples were detected negative by this method). This result was independent of the used qPCR system. The Ct values from the three extraction procedures amplified with the four different qPCR assays are presented in Table 3 (mean Ct values) and Appendix A. In general, differences could be shown for the extraction methods only. All qPCR assays, regardless of whether lyophilized or not, delivered very similar results. While showing the lowest Ct values, the lyophilized IndiField ASFV PCR was also not able to detect the two borderline samples, 3 and 11 (Appendix A), extracted with the TLR method that were also not detected by the nonlyophilized qPCR assays.

## 3. Discussion

African swine fever has triggered global concerns; highly significant economic impact and mortality rates have led to a major threat to the pig industry. Without ASF-specific treatment or an effective vaccine, rapid and accurate laboratory diagnosis is an important tool for timely intervention and thus ASF control. The actual lab diagnosis focuses on viral nucleic acid isolation and PCR from available specimens and antibody detection from liquid samples [15]. Molecular diagnostic techniques in the EU reference laboratories are mainly based on OIE-recommended methods (i.e., conventional [10] and real-time PCR systems [3,7,9,12,13,16] and several commercial ASFV real-time PCR kits).

In this study, seven nucleic acid extraction methods and four different real-time PCR assays for ASFV detection were compared. Different sample materials were used and collected from several animal experiments with strains of different genotypes. The results showed that a simplification of this kind of assays and workflows can be achieved with no relevant loss of sensitivity or specificity. This should encourage the use of its broad application in different labs. 

The data analyses for matrix selection confirmed that EDTA blood is the most suitable choice for ASFV genome detection in both initial and late phases of infection of live animals. This result correlates with the work of other groups [9,13,17]. Serum samples could be also detected in the early stage of infection, but with a considerably reduced viral genome load in comparison with EDTA blood. Alternative specimens, like oral swabs or chewing ropes, could detect ASFV to a certain extent in the late phase of infection based on the increased viremia with significantly lower genome loads. For postmortem analyses, we could confirm that spleen is the most appropriate material for ASFV detection. This result was consistent with similar investigations [14]. In general, EDTA blood or bloody tissue materials are recommended for ASFV detection from both experimentally infected animals and dead carcasses in the field.

All tested silica membrane- or magnetic bead-based extraction methods were comparatively sensitive for DNA isolation. The manual QIAamp Viral RNA Mini Kit (Qiagen), based on the silica membrane system, could successfully isolate the viral DNA of the ASFV genome. Similar results could be ascertained in a study of Haines et al. [13]. The authors could also demonstrate that this kit is convenient for viral DNA extraction from ASFV. Additionally, our study showed that this kit could deliver almost identical results in regard to Ct values compared with automated magnetic bead-based extraction methods. For the automated magnetic bead-based systems, no differences between the usage of different input sample volumes or prefilled or non-prefilled reagents and different instruments (KingFisher Flex System or IndiMag48) could be observed. However, prefilled reagents have the ability of being conducted on both automated systems. The advantage of the IndiMag48 instrument is the possibility to extract nearly all exact sample numbers between 1 and 48 based on the individual composition of plasticware for 1, 8, and 24 samples. On the other hand, the KingFisher system has a wide range of extraction of up to 96 samples simultaneously, which could be perfectly practical in case of high-throughput scenarios as free testing of swine populations in ASFV restriction zones. 

The virotype Tissue Lysis Reagent (TLR) was developed for the fast preparation of various sample types without the need for an extraction kit or any complicated nucleic acid isolation procedures and has been successfully used for Bovine viral diarrhea virus (BVDV) diagnosis from ear notch samples [18]. The viral ASFV genome release by TLR showed a slightly lower sensitivity compared with the standard silica membrane- and magnetic bead-based systems. However, the TLR could have the advantage of a successful application in a wide range of diagnostic laboratories in case of limited or unavailable commercial extraction kits or reagents. Especially, the COVID-19 pandemic situation has generated a huge consumption of extraction kits, and thus, the TLR method could be an effective alternative for the continuation of molecular ASFV diagnostics. For high-throughput scenarios, up to 96 samples can be processed with the TLR in appropriate PCR plates. The incubation can be performed in a conventional PCR thermocycler, followed by centrifugation in a plate centrifuge (e.g., 5804 R centrifuge, Eppendorf, Hamburg, Germany).

The four tested real-time PCR assays could detect the ASFV genome with similar efficiency. The most sensitive PCR was obtained from the IndiField ASFV PCR, which was amplified on the IndiField thermocycler. The study of Daigle et al. (2020) has ascertained the functionality of the IndiField thermocycler [19]. The slightly increased analytical sensitivity of the IndiField ASFV PCR compared with the other tested PCR assays can be most likely explained by the high template volume, possibly due to the lyophilized format of the kit. Interestingly, the IndiField ASFV PCR delivered excellent PCR result in a short time using a temperature profile of less than 60 min. The other three PCR assays with their liquid chemistry could achieve comparable results with high sensitivity and efficiency, which were conducted on a standard real-time thermocycler (Bio-Rad CFX96). The liquid master mixes can be used on different real-time PCR thermocyclers. However, it was not suitable to analyze the complete test panel on the IndiField thermocycler due to its limitation of up to nine samples per run. Nevertheless, the use of lyophilized ready-to-use reagents and the related higher template input, as well as master mix stability, may in the future also be available for standard real-time PCR platforms if appropriate plastic is used. A previous study successfully demonstrated that ASFV could be detected by the use of lyophilized reagents for qPCR amplification [20]. Here, prefilled single tubes, 8-well strips, 24-well blocks, and complete 96-well plates can be used for the individual application of cycler-specific PCR kits. In general, the application of prefilled (lyophilized) pathogen-specific PCR kits would be an excellent extension of the use of prefilled reagents for nucleic acid extraction and would further reduce the risk of contaminations and the working time in the molecular diagnostic procedures. This was correlated to the works of other groups, which were performed with different pathogens, such as the influenza A virus [21] and bluetongue virus [22].

The standard and “easy lab” methods were successfully applied for ASFV detection in field specimens collected from dead wild boars during the 2020 ASFV outbreak in Germany. The data showed that all methods not only are convenient for samples from live animals but also can be successfully applied for different sample materials from carcasses of wild boars.

## 4. Materials and Methods

### 4.1. Sample Collection from Experimentally Infected Animals

A panel consisted of 90 samples from domestic pigs and wild boar that had been obtained in seven different animal experiments with ASFV strains of different genotypes (Table 4). The animal trials were approved by a competent authority (Landesamt für Landwirtschaft, Lebensmittelsicherheit und Fischerei (LALLF) Mecklenburg-Vorpommern, Rostock, Germany) under reference number 7221.3-2.011/19. Different samples of these animal trials were used for the validation study (EDTA blood, serum, oral swabs, tissue homogenate spleen samples, and chewing ropes collected at different time points post-infection). In summary, 36 EDTA blood samples, 25 serum samples, 20 oral swabs, 6 chewing ropes, and 3 tissue homogenate spleen samples were used in this study (details shown in Appendix A). The animals were housed in groups in the high containment facility of the Friedrich-Loeffler-Institut (FLI) (L3^+^). The animals were fed a commercial pig food with corn and hay cob supplement and had access to water ad libitum.

EDTA blood and serum samples were collected by using the KABEVETTE® G system (KABE Labortechnik, Nümbrecht, Germany). Afterwards, blood samples were prepared for long-term storage at +4 °C by adding penicillin/streptomycin, 100× (Thermo Fisher, Darmstadt, Germany) and gentamicin/amphotericin B solution, 500× (Thermo Fisher, Darmstadt, Germany), while serum samples were centrifuged at 4000 rpm for 10 min. Finally, both sample types were stored at +4 °C until the DNA extraction step. An amount of 0.5 g of organ tissue samples was homogenized by grinding with a 5 mm steel ball within 1 mL cell culture medium in 2 mL bolted tubes using the TissueLyser II (Qiagen, Hilden, Germany).

Additionally, oral swabs (Copan Diagnostics Inc., Brescia, Italy) from individual pigs and chewing ropes from each stable were used for noninvasive sample collection. Oral swabs and pieces of chewing rope samples were enriched in 2 mL standard cell culture medium including antibiotics (see above) and incubated at room temperature on a thermoshaker (VWR International GmbH, Darmstadt, Germany) for 30 min (oral swabs) or 24 h (chewing ropes). The supernatant was used for the DNA extraction procedures.

### 4.2. Field Samples from ASFV Outbreak in Germany 

Different specimens with sufficient sample volume collected from the first ASFV outbreaks in September 2020 in Germany were used for the evaluation. The samples delivered from the State Laboratory Berlin-Brandenburg were collected from carcasses found in the border region to Poland. A total of 14 samples (serum, bone marrow, and bloody swab suspensions) from 14 different wild boars were selected for the investigations. This panel consisted of 10 swab suspensions, 1 serum sample, and 3 bone marrow homogenates (gathered in Appendix A).

Swab suspension was generated in 1.5 mL cell culture medium; the serum samples were centrifuged at 8000 rpm for 1 min before use. Bone marrow samples were homogenized by grinding 0.5 g of organ tissue with a 5 mm steel ball within 1 mL phosphate-buffered saline in 2 mL bolted tubes. 

### 4.3. DNA Extraction 

Seven different extraction and releasing methods were applied for the ASFV DNA isolation.

QIAamp Viral RNA Mini Kit (Qiagen, Hilden, Germany): This silica membrane-based extraction kit is well established and is widely used for the manual extraction of both DNA and RNA from cell-free and cell-containing specimens. Briefly, a reduced sample volume of 70 µL to avoid the overload of the silica membrane was mixed with 560 µL AVL lysis buffer of the kit. An amount of 5 µL of internal control DNA (IC2-DNA) [23] was added to the sample–lysis buffer mixture, vortexed, and incubated at room temperature for 10 min. The following steps of the extraction procedure are based on the manufacturer’s instructions. Finally, the nucleic acid was eluted in 50 µL elution buffer and stored at −20 °C. Using this kit, DNA/RNA for up to 12 samples can be extracted in approximately 30 min.NucleoMagVet Kit (Macherey-Nagel, Düren, Germany): This magnetic bead-based extraction kit was conducted on the KingFisher Flex System (Thermo Fisher Scientific, Darmstadt, Germany). Briefly, 100 µL sample volume was added to 100 µL VL1 lysis buffer and processed according to the instructions of the manufacturer. For internal control, 10 µL IC-DNA was mixed with 350 µL VEB binding buffer per sample and was added to the sample–lysis buffer mixture. After three washing steps, the extracted nucleic acid was eluted in 100 µL elution buffer. The extraction protocol on the KingFisher Flex System needs approximately 20 min for up to 96 samples. Details of the KingFisher protocol can be provided on request.NucleoMagVet- Kit (Macherey-Nagel) on the KingFisher Flex System, which was performed identically with the same protocol as described above in B, however, it was used with a different sample input volume of 200 µL.IndiMag^®^ Pathogen Kit: This magnetic bead-based extraction kit was applied on the IndiMag48 instrument (both kit and machine from Indical Bioscience, Leipzig, Germany). An interesting highlight of the IndiMag48 instrument is the variability of the number of extraction samples, which can be performed per run. Plastic blocks for 1, 8, or 24 samples can be combined to cover nearly all numbers between 1 and 48 samples. For each sample, four wells were used for the extraction procedure. Briefly, in the first well, 20 µL proteinase K was mixed with 200 µL sample and 500 µL VXL mixture (100 µL VXL lysis buffer, 400 µL ACB binding buffer, 25 µL magnetic beads, and 10 µL IC-DNA). In the second and third wells, the AW1 buffer (wash 1) and the AW2 buffer (wash 2) were housed, respectively. Finally, the nucleic acid was eluted in 100 µL elution buffer. The extraction procedure was realized according to the manufacturer’s instructions, and the extraction time for up to 48 samples on the IndiMag48 platform was 31 min.IndiMag^®^ Pathogen IM48 Cartridge (IndiMag^®^ Pathogen Kit prefilled for the IndiMag48 instrument): Here, the different buffers were prefilled into the four wells used per sample for the extraction. In the first well, the 20 µL proteinase K and, in the second well, the AW1 buffer mixed with magnetic beads were present. The AW2 buffer and the elution buffer were prefilled in wells 3 and 4, respectively. The prefilled and sealed plates were produced by Indical Bioscience and used according to the manufacturer’s instructions. An amount of 200 µL sample volume, 500 µL VXL/ACB mixture without magnetic beads, and 10 µL IC-DNA (supplied with the virotype ASFV 2.0 PCR Kit) were added in the first well and then conducted directly on the IndiMag48 instrument with the same protocol used as for the non-prefilled extractions.IndiMag^®^ Pathogen KF96 Cartridge (IndiMag^®^ Pathogen Kit prefilled for the KingFisher Flex System): Here, five prefilled 96 deep-well plates were provided by Indical Bioscience (plate 1 = proteinase K, plate 2 = AW1 buffer mixed with magnetic beads, plate 3 = AW2 buffer, plate 4 = AW3 buffer (supplementary wash step), and plate 5 = elution buffer). For the extraction, 200 µL sample, 500 µL VXL/ACB mixture without magnetic beads, and 10 µL IC-DNA (supplied with the virotype ASFV 2.0 PCR Kit) were added into the wells of the first plate. Extraction time was 32 min.Nucleic acid release method of the ASFV genome by virotype Tissue Lysis Reagent (TLR) from Indical Bioscience: Here in this study, 10 µL ASFV sample was added to 90 µL TLR buffer in a standard 1.5 mL Eppendorf tube and mixed very well by pipetting up and down. The sample–TLR buffer mixture was incubated at 65 °C for 30 min and at 98 °C for 15 min, followed by cooling to room temperature. Afterwards, the sample–TLR buffer mix was centrifuged at 10 000 ×*g* for 10 min. Finally, the cleared supernatant was transferred directly into the PCR reaction tube as template.

In all the extraction procedures, two exogenous extraction control DNAs were added to all lysis buffers of each extraction method (enhanced green fluorescent protein gene mix [23] and IC-DNA from the virotype ASFV 2.0 PCR Kit) according the references. The extracted template nucleic acids were stored at −20 °C until use.

### 4.4. Real-Time PCR Kits and Assays for ASFV Detection

Four different qPCR assays for ASFV genome detection were comparatively tested:Haines PCR: The PCR assay described by Haines et al. [13] was modified by using a lab-specific amplification mix and the integration of a lab-specific internal control system utilizing the PerfeCTa® qPCR ToughMix® Kit from Quanta BioSciences (Gaithersburg, MD, USA). A FAM-labelled ASFV primer–probe mixture consisted of 800 nM ASFV-p72IVI-F, 800 nM ASFV-p72IVI, and 200 nM ASFV-p72IVI probe in 0.1 × TE buffer (pH 8.0). For the control of extraction and qPCR amplification, a heterologous control system, published by Hoffmann et al. [23], was integrated. Here, a HEX-labelled primer–probe mixture consisted of 200 nM EGFP1-F, 200 nM EGFP2-R, and 200 nM EGFP probe 1 in 0.1 × TE buffer (pH 8.0). The 12.5 µL total reaction mix was established by 1.75 µL RNase-free water, 6.25 µL 2× PerfeCTa qPCR ToughMix, 1.0 µL ASFV primer–probe mix (ASFV-P72-IVI-Mix-FAM), 1.0 µL internal control primer–probe mix (EGFP-Mix1-HEX), and 2.5 µL DNA template. The following thermoprofile was used for amplification: 3 min at 95 °C, 45 cycles at 95 °C for 15 s, 60 °C for 20 s, and 75 °C for 20 s. The fluorescence data in the FAM and HEX channel were collected during the annealing step, and the total run time on the Bio-Rad CFX96 Real-Time Detection System (Bio-Rad, Hercules, CA, USA) was 1 h and 16 min. For the data analyses, the Bio-Rad Maestro software (version 4. 1.2433. 1219) was used.EURL PCR: This method is recommended by the EU reference lab for ASF and based on the publication of Fernández-Pinero et al. [9]. The qPCR is listed as the official method by the OIE. Because the original UPL probe is not commercially available anymore, an alternative TaqMan probe was introduced by the EURL-ASF. In our tests, the LightCycler 480 Probes Master Kit (Roche Applied Science, Mannheim, Germany) was used for the amplification according the standard operating procedure on the website of the EURL-ASF (https://asf-referencelab.info/asf/en/procedures-diagnosis/sops, accessed on 25 January 2021). Briefly, FAM-labelled ASF-VP72 primer–probe mixtures consisted of 600 nM ASF-VP72-F, 600 nM ASF-VP72-R, and 200 nM ASF-VP72P1-FAM in 0.1 × TE buffer (pH 8.0). For the internal control amplification, the EGFP-Mix1-HEX, as described above, was used. A total reaction PCR mix of 20 µL volume containing 6.0 µL RNase-free water, 10.0 µL of 2× LC480 Probes Master PCR Mix, 1.0 µL ASF-VP72-Mix-FAM, 1.0 µL EGFP-Mix1-HEX, and 2.0 µL template DNA was prepared. The PCR conditions were 5 min at 95 °C, followed by 45 cycles at 95 °C for 10 s and 60 °C for 30 s. The fluorescence data in the FAM and HEX channel were collected during the annealing step, and the total run time on the CFX96 Real-Time Detection System was 1 h and 13 min.Virotype ASFV 2.0 PCR Kit (Indical Bioscience, Leipzig, Germany): This qPCR assay is a commercial kit for the detection of ASFV and is licensed for the German market. An amount of 20 µL of the ready-to-use master mix was filled in the PCR reaction well, and 5 µL of the template DNA was added to give a final reaction volume of 25 µL. Besides the ASFV target amplification, the master mix features two independent control systems. The homologous (endogenous) extraction and amplification control is detected in the HEX/JOE channel, whereas an additional heterologous (exogenous) extraction control is detected in the Cy5 channel. The exogenous control (IC-DNA) is supplied with the virotype ASFV 2.0 PCR Kit and is added to the lysis buffer during extraction. These controls serve to control extraction from the animal sample and to identify samples showing full and partial inhibition, thus excluding false-negative ASFV samples. According the supplier’s instructions, a run time of 59 min on the CFX96 Real-Time Detection System with the following temperature profile was conducted: 2 min at 95 °C, 40 cycles at 95 °C for 5 s, and 60 °C for 30 s [14].IndiField ASFV PCR (Indical Bioscience, Leipzig, Germany): This commercial real-time PCR amplifies the ASFV genome in the FAM channel and a homologous internal extraction control in the Amber/Texas Red channel. Interestingly, the PCR reactions were prepared as ready-to-use lyophilized reagents in the individual PCR tubes of the ultraportable IndiField thermocycler. The reaction mix was prepared by adding 20 µL DNA template directly to the lyophilized master mix. The cycler is fully controlled by a smartphone, and up to nine samples in one run can be analyzed in parallel. The PCR data can be uploaded to a cloud-based storage and analysis system. A PCR thermoprofile of 1 min at 95 °C, followed by 45 cycles at 95 °C for 1 s and 60 °C for 20 sec, will be introduced by scanning the specific QR code on the package of the lyophilized IndiField ASFV PCR. The total run time for this system on the IndiField thermocycler is 56 min.

Dilution series of an ASFV DNA standard (ASFV Estonia 2014) were applied in each PCR run to confirm the sensitivity and reproducibility of the performed analyses (Appendix A).

### 4.5. Statistical Analysis

Initial data recording and analyses (comparison of mean values and transformation of values) were done using Microsoft Excel 2010 (Microsoft Germany GmbH, Munich, Germany). GraphPad Prism 8 (GraphPad Software Inc., San Diego, CA, USA) was used for further statistical analyses and graph creation. Statistically significant differences were investigated by two statistical tests (unpaired *t*-test and one-way ANOVA) to test the significance of the results. Statistical significance was defined as *p* < 0.05 and indicated with an asterisk (*); *p* < 0.01 was indicated with two asterisks (**).

## 5. Conclusions

EDTA blood and bloody materials are the sample matrices of choice for a sensitive ASFV genome detection, independent of the course and phase of the disease. Serum samples also work fine in general, but here, the sensitivity of the DNA detection in the early phase of infection can be reduced. Noninvasive sample materials (oral swabs and chewing ropes) are clearly less suitable for the detection based on the minimal virus excretion.

If the optimal specimens are used for the molecular detection of ASFV, several extraction and qPCR methods are “fit for purpose.” The selection of ideal systems for a specific lab depends on various factors. To name a few, the number of analyses per day, the available lab equipment, the budget, the personal and technical resources, and the necessity to use certified kits are of relevance. Depending on the specific situation in the lab, the different methods for extraction and qPCR presented here can be combined in a modular regime. In addition, viral DNA release via the TLR procedure can be an option in the molecular diagnostics of ASFV, especially if standard extraction kits are expensive or not available. 

In our study, we could show that simplification of DNA extraction and qPCR does not result in reduced diagnostic sensitivity per se. Based on the minimization of manual handling and working time, the use of commercially available and prefilled reagents for extraction and qPCR can reduce the risk of false-negative and false-positive results especially in high-throughput scenarios. The implementation of state-of-the-art internal control systems and easy-to-handle software in the used machines, combined with improved storage stability by using lyophilized PCR kits, will further improve the diagnostic safety and robustness of molecular diagnostics.

## Figures and Tables

**Figure 1 ijms-22-02307-f001:**
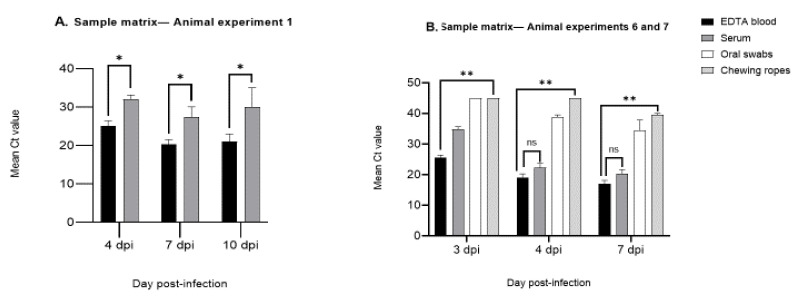
(**A**) Sample matrix comparison (EDTA blood and serum) from animal experiment 1. The mean Ct values based on five live domestic pigs (animal numbers 30, 31, 32, 35, and 37) inoculated with African swine fever virus (ASFV) Estonia 2014 at different time points, 4, 7, and 10 dpi (number of replicates = 7, Appendix A) are shown. Standard deviation (SD) for EDTA blood (2.68) and serum (1.44). An unpaired *t*-test was performed for statistical analysis, and EDTA blood showed significantly lower Ct values among the different time points, 4, 7, and 10 dpi (* *p*-value < 0.01, number of replicates = 7, Appendix A). (**B**) Sample matrix comparison (EDTA blood, serum, and oral swabs) and two animal pens (chewing ropes) from animal experiments 6 and 7, mean Ct values based on four live domestic pigs (animal numbers 48, 51, 53, and 58) inoculated with two different ASFV strains (KAB 6/2 and SUM 14/11) at different time points, 3, 4, 7, and 8 dpi. SD values for EDTA blood (4.45), serum (7.90), oral swabs (5.27), and chewing ropes (3.20). Comparing the overall genome loads, an unpaired *t*-test was performed to test the significance of each matrix. EDTA blood showed highly significant Ct values compared with other matrix samples at 3 dpi (** *p*-value = 0.002). A similar significance level could be identified for oral swabs (** *p*-value = 0.009) and for chewing ropes (** *p*-value = 0.002). However, at 4 and 7 dpi, the ASFV genome load in serum was not significantly different from the genome load in EDTA blood (ns *p*-value = 0.3).

**Figure 2 ijms-22-02307-f002:**
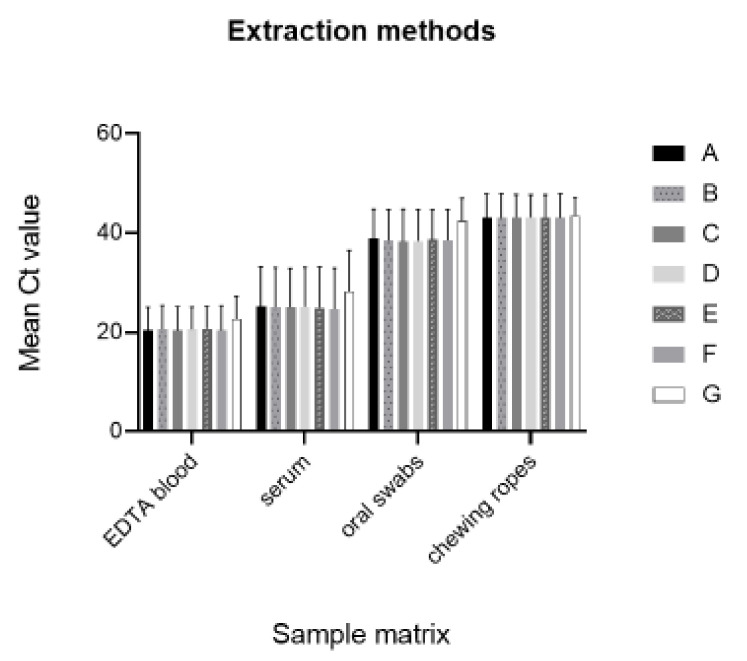
Extraction method comparison, mean Ct values obtained from 30 animals (EDTA blood), 25 animals (serum), 20 animals (oral swabs), and 6 animal pens (chewing ropes). (**A**) QIAamp Viral RNA Mini Kit (70 µL sample volume). (**B**) NucleoMagVet Kit (100 µL sample volume). (**C**) NucleoMagVet kit (200 µL sample volume). (**D**) IndiMag^®^ Pathogen Kit. (**E**) IndiMag^®^ Pathogen IM48 Cartridge. (**F**) IndiMag^®^ Pathogen KF96 Cartridge. (**G**) Nucleic acid release by virotype TLR. (Sample volume for all IndiMag^®^ extraction formats was 200 µL). SD analysis was carried out (number of replicates = 30); for mean Ct values, see Table 1. SD value for **A**, 10.79; **B**, 10.68; **C**, 10.75; **D**, 10.68; **E**, 10.76; **F**, 10.86; and **G** (10.41). Standard error of the mean value for **A** is 5.39; **B**, 5.34; **C**, 5.37; **D**, 5.34; **E**, 5.37; **F**, 5.42; and **G**, 5.20. A one-way ANOVA was performed to test the significance between the different extraction methods based on the same matrix samples with a resulting *p*-value > 0.99 for the taken samples, which is not statistically significant.

**Figure 3 ijms-22-02307-f003:**
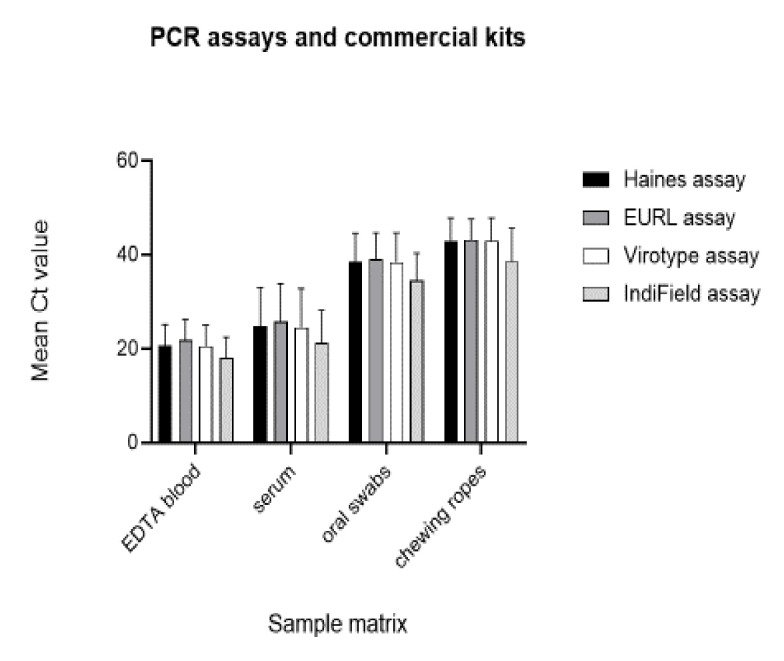
Comparison of PCR assays and commercial kits, mean Ct values obtained from 30 animals (EDTA blood), 25 animals (serum), 20 animals (oral swabs), and 6 animal pens (chewing ropes). (1) PerfeCTa qPCR ToughMix Kit (Haines assay). (2) LightCycler 480 Probes Master Kit (EURL assay). (3) Virotype ASFV 2.0 PCR Kit (virotype assay). (4) IndiField ASFV PCR (IndiField assay). SD analysis was carried out (number of replicates = 30); for mean Ct values, see Table 2. SD values for Haines assay, 10.07; EURL assay, 10.25; virotype assay, 10.77; and IndiField assay, 10.02. Standard error of the mean value for Haines assay is 5.34; EURL assay, 5.12; virotype assay, 5.38; and IndiField assay, 5.00. A one-way ANOVA was performed to test the significance between the different PCR assays based on the same matrix samples with a resulting *p*-value = 0.93 for the taken samples, which is not statistically significant.

**Table 1 ijms-22-02307-t001:** Mean Ct values of different DNA extraction methods using different sample materials.

Sample Matrix	Extraction Methods *						
	A	B	C	D	E	F	G
EDTA blood	20.44	20.61	20.04	20.64	20.71	20.38	22.67
Serum	25.22	24.98	24.96	25.09	24.88	24.63	28.22
Oral swabs	38.92	38.47	38.26	38.43	38.68	38.55	42.49
Chewing ropes	43.05	43.03	43.11	43.15	43.15	43.03	43.58

* For description of the extraction methods, use the legend of Figure 2.

**Table 2 ijms-22-02307-t002:** Mean Ct values of ASFV qPCR assays using different sample materials.

Sample Matrix	Total Sample Number	Haines Assay Mean Ct Value (pos. Sample No.)	EURL Assay Mean Ct Value (pos. Sample No.)	Virotype Assay Mean Ct Value (pos. Sample No.)	IndiField Assay Mean Ct Value (pos. Sample No.)
EDTA blood	36	20.7 (34)	21.8 (34)	20.6 (34)	18.1 (35)
Serum	25	24.8 (23)	25.8 (23)	24.5 (23)	21.2 (25)
Oral swabs	20	38.6 (11)	39.1 (11)	38.4 (11)	34.5 (16)
Chewing ropes	6	43.0 (1)	43.2 (1)	43.0 (1)	38.7 (3)

**Table 3 ijms-22-02307-t003:** Testing of ASFV-positive field samples from the outbreak 2020 in Germany. The Ct values of three different extraction methods and four different ASFV qPCR assays are shown.

Animal	Sample Matrix	(1) Haines Assay			(2) EURL Assay			(3) Virotype Assay			(4) IndiField Assay		
		N	I	T	N	I	T	N	I	T	N	I	T
		Ct Value			Ct Value			Ct Value			Ct Value		
1	SwS	30.7	30.4	35.9	31.3	31.2	36.4	29.3	28.9	35.1	25.8	25.8	38.6
2	SwS	27.6	28.1	31.4	28.1	28.8	32.4	26.0	26.6	29.8	22.7	23.1	28.2
3	SwS	31.9	31.7	-	32.7	32.7	-	30.3	30.6	-	27.1	26.8	-
4	SwS	25.4	25.3	30.9	26.0	26.0	32.1	23.7	23.7	29.1	19.8	20.2	26.8
5	Serum	29.9	30.2	31.1	29.5	30.0	31.3	29.0	28.3	29.3	25.2	25.0	27.3
6	SwS	28.1	28.0	30.9	28.3	28.7	31.9	26.3	26.3	29.2	23.0	23.0	27.7
7	SwS	29.6	30.4	35.6	30.1	30.7	36.7	27.9	28.7	36.5	24.8	24.9	32.1
8	SwS	20.5	20.6	23.6	21.1	21.5	24.6	19.2	19.3	22.5	16.0	16.0	25.2
9	BM	21.9	21.4	26.7	22.1	22.1	28.0	20.5	19.8	27.8	17.0	17.0	25.8
10	BM	18.8	18.5	21.5	19.2	19.1	22.6	17.3	17.2	20.4	14.1	15.1	22.6
11	BM	34.5	35.0	-	36.3	36.9	-	33.1	34.5	-	30.7	31.6	-
12	SwS	26.1	25.0	29.1	26.1	26.0	29.9	24.1	23.9	27.4	20.0	19.8	25.9
13	SwS	22.7	22.1	25.4	23.0	22.6	26.8	21.1	20.3	24.1	17.0	16.9	23.9
14	SwS	27.8	27.6	30.1	27.9	28.0	31.3	26.0	25.8	28.8	22.1	22.1	26.9
15	DIC	-	-	-	-	-	-	-	-	-	-	-	-
16	DIC	-	-	-	-	-	-	-	-	-	-	-	-

(**1**) PCR assay based on the protocol published by Haines [13]. (2) EURL PCR assay, which is an OIE-recommended method [9]. (**3**) Virotype ASFV 2.0 PCR Kit [14]. (**4**) IndiField ASFV PCR. Abbreviations: N = NucleoMagVet kit (Macherey-Nagel), I = IndiMag^®^ Pathogen Kit (Indical Bioscience), T = Tissue Lysis Reagent (Indical Bioscience), SwS = swab suspension, BM = bone marrow, DIC = DNA isolation control (ASFV negative serum), - = no Ct.

**Table 4 ijms-22-02307-t004:** African swine fever virus isolates used in this study. Abbreviations: o.-n. = oro-nasally; i.m. = intramuscularly; HAD = hemadsorbing doses.

Animal Experiment	Genotype	Isolate	Country of Origin	Year	Infection Route	Infection dose (HAD50 /mL)
1	II	Estonia 2014	Estonia	2014	o.-n.	10^5.25^
2	IV	RSA W1/99	South Africa	1999	i.m.	10^0.83^
3	XII	MFUE 6/1	Zambia	1982	i.m.	10^1.16^
4	XIX	CHZT 90/1	Zimbabwe	1990	i.m.	10^1.0^
5	II	Belgium 2018/1	Belgium	2018	o.-n.	10^4.6^
6	XI	KAB 6/2	Zambia	1983	i.m.	10^3.25^
7	XIII	SUM 14/11	Zambia	1983	i.m.	10^3.3^

## Data Availability

The data set used and/or analyzed during the current study are available from the corresponding author on reasonable request.

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
