# Peer review of "Swift and Reliable “Easy Lab” Methods for the Sensitive Molecular Detection of African Swine Fever Virus"

_ijms, 2021, doi:10.3390/ijms22052307_

Round 1
Reviewer 1 Report
African swine fever (ASF) is a severe contagious viral hemorrhagic disease for pigs, which pose a great threat for swine industry and have results in huge economic losses. African swine fever. The main strategy for controlling the ASF is rapid diagnosis and eradication program since no vaccine is available now. The manuscript by Elnagar et al. established the “easy lab” strategy for the molecular detection of African swine fever virus. This optimized method would be helpful for the rapid and accurate diagnosis of ASFV.
However, the authors should further improve the manuscript before it can be considered to be published:
1.In Page 3 and 4, why not perform the SD analysis in Figure 1 and Figure 2?
2 .All tables (Table 1, 2, 3) seems not to be standard three-wire grid. In table 3, where is “experiment 1”?
3. In Line 231 and 368, what is “ASPV”?
4. In Line 420, Please check whether “…will be….” is OK or not?
5. The authors should describe the biosafety condition of their lab to perform these experiments.
Reviewer 2 Report
Authors have extensively work on the comparison of several DNA isolation methods, from different sample matrix specimenes with several different qPCR assays, in order to give several alternatives for the detection of ASFV DNA in live and post-morten animals. I really consider this article important for the scientific community, focused on a relevant animal health problem. Introduction is complete and clear, and the experiments are generally well designed and I think that it could be suitable for Int J. of Molecular Sciences. However, I have some comments and/or modification requests before accepting this paper. The most important concern, from my point of view, is the lack of a statistical analysis, that I consider essential, specially for a study with many analysis of comparison.
With regards to Material and methods:
Table 3: Animal experiment info, in first line of the table, is missing.
Line 279: tamicin/Amphotericin B solution (500x, ThermoFisher). While serum samples were cen [...]. Substitution of the end point by a comma should be done.
Line 281-283: Tissue samples were homogenized by grinding approximately 0.5 gram of organ tissue with 5 mm steel ball within 1ml cell culture medium in 2ml bolted tubes using the TissueLyser II. I consider that it might be more correct to say: 0.5 gram of organ tissue samples were homogenized by grinding with 5 mm steel [...]
Line 325: as described in (B). This reviewer can not easily identify what the authors meant with (B).
Line 374: -p72IVI- instead of -p72IV-V
Line 375: please, substitute "und" by "and".
Line 402: virotype ASFV 2.0 PCR Kit (Indical Bioscience, Leipzig, Germany): This qPCR assay --> please, check upper and lower-case letters in this sentence. Same for line 416.
Regarding Results section:
This section is generally well structured. However, it is a bit confussing because of the lack of information in order to make easier for the reader to follow the article. Thus:
Fig. 1A and B are not enough informative, graphs or caption should include information about the number of replica and standard error/deviation of the data, even although some of this info can be find in Sup. material. Moreover, this reviewer has some other questions/comments about this fig:
- Why Fig 1A shows only 2 matrix samples from animal experiment 1, why is "Spleen" out of this figure (according to Table S1)? I guess that it is because authors are describing matrix from live animals, but it could be indicated again along the paragraph/caption.
- Line 92, in caption: mean Ct values based on five : Five Animals? I suppose that authors refer to animals 30,31, 32, 35 and 37 (according to Table S1), but they should make easier to find this info! Please, add (Table S1) in the results section to lead the reader to that data.
*Note: Colors of the different columns are quite similar among them, except for the black one, I suggest authors to change one of the grey ones by white.
Lane 85-86: Serum samples delivered comparable results to EDTA blood, but only in the later stage of infection. Authors refered to Fig 1B? Has this comparison been statistically analysed? Please, provide this information.
The same comment for the rest of the sections. I strongly ask for statistical analysis and basic statistical information for this article (mean Ct ±SD, p-value, N value, number of replica) since it is a comparison study.
Please, check format of Table 2.
Lines 455-457: only information about Supplementary material should be included in this section.
Reviewer 3 Report
The manuscript is spotless and authors demonstrate their unique expertise in the field
Despite I am not a specialist in diagnostics, the manuscript reads easily
As a minor criticism, it was not easy to discern the very novel and original results
Round 2
Reviewer 1 Report
The authors have modified the manuscript accordingly, I have no additional comment.